# Estimation of Interaction Locations in Super Cryogenic Dark Matter Search Detectors Using Genetic Programming-Symbolic Regression Method

Nikola Anđelić \*,†, Sandi Baressi Šegota †, Matko Glučina and Zlatan Car

Department of Automation and Electronics, Faculty of Engineering, University of Rijeka, Vukovarska 58, 51000 Rijeka, Croatia
\* Correspondence: nandelic@riteh.hr
† These authors contributed equally to this work.

**Abstract:** The Super Cryogenic Dark Matter Search (SuperCDMS) experiment is used to search for Weakly Interacting Massive Particles (WIMPs)—candidates for dark matter particles. In this experiment, the WIMPs interact with nuclei in the detector; however, there are many other interactions (background interactions). To separate background interactions from the signal, it is necessary to measure the interaction energy and to reconstruct the location of the interaction between WIMPs and the nuclei. In recent years, some research papers have been investigating the reconstruction of interaction locations using artificial intelligence (AI) methods. In this paper, a genetic programming-symbolic regression (GPSR), with randomly tuned hyperparameters cross-validated via a five-fold procedure, was applied to the SuperCDMS experiment to estimate the interaction locations with high accuracy. To measure the estimation accuracy of obtaining the SEs, the mean and standard deviation ($\sigma$) values of $R^2$, the root-mean-squared error ($RMSE$), and finally, the mean absolute error ($MAE$) were used. The investigation showed that using GPSR, SEs can be obtained that estimate the interaction locations with high accuracy. To improve the solution, the five best SEs were combined from the three best cases. The results demonstrated that a very high estimation accuracy can be achieved with the proposed methodology.

**Keywords:** cross-validation; genetic programming; interaction location; SuperCDMS; symbolic regression





## 1. Introduction

The Cryogenic Dark Matter Search (CDMS) can be described as a series of specially designed experiments that are used to detect Weakly Interacting Massive Particles (WIMPs), i.e., dark matter, using an array of semiconductor detectors at extremely low temperatures (mK). The exact definition of WIMPs does not exist; however, they are described as hypothetical new elementary particles (candidates for dark matter) that interact via gravity and other forces and are not part of the Standard Model. The product of every particle interaction in the germanium and silicon substrate produces ionization and phonons, which are measured using CDMS detectors [1]. The measurement of ionization and phonons determines the energy deposited in the crystal for each interaction and provides information on which kind of particle caused the event. Every particle's interaction with atomic electrons (electron recoils) and atomic nuclei (nuclear recoils) results in different ratios of ionization and phonon signals. The majority of particle interactions are electron recoils, while WIMPs are expected to be nuclear recoils. Although the WIMP scattering events are unique, they are rare when compared to the vast majority of unwanted interactions.

Two types of experiments exist, i.e., CDMS and SuperCDMS. The CDMS experiment provided the most-sensitive test of potential WIMP–nucleon interactions, as reported

in [2,3]. The XENON experiment [4] later surpassed the CDMS experiment. The Super-CDMS experiment [5] is the successor of the CDMS, which uses new and improved detectors and an increased target mass to improve sensitivity by a factor of 5–8 factor and is limited by the residual cosmogenic background at the current location.

One of the challenges in CDMS and SuperCDMS experiments is the detection accuracy of particle interactions and the reconstruction of particle interaction locations, which could be greatly improved with the use of machine learning (ML) algorithms. In recent years, scientists have implemented various ML algorithms to estimate or detect dark matter using data obtained from CDMS/SuperCDMS experiments. Some of these research papers are briefly described in the following subsection.

### 1.1. Application of ML Algorithms in CDSM/SuperCDSM Experiments

A semi-supervised ML approach to dark matter search was proposed in [6]. This paper used a convectional auto-encoder and semi-supervised convolutional neural network (CNN) to directly detect dark matter. A deep convolutional neural network (DCNN) was used in [7] to solve the tasks of retrieving Lagrangian patches from which dark matter halos will condense. The results of the investigation showed that, if the proposed methods were properly tuned, they can outperform likelihood-based methods. The recurrent neural network (RNN) was implemented on the trigger FPGA to maximize the sensitivity to the low-mass dark matter of the SuperCDMS SNOLAB experiment [8]. By doing so, the energy estimator based on the combined information of filtered traces from individual detector channels was improved. With the performed modifications, the trigger threshold was lowered by 22%. The deep learning method was proposed in [9] to map the 3D galaxy distribution in hydrodynamic simulations and the underlying dark matter distribution. In this research, as two-phase CNN was used to generate fast galaxy catalogs, and the results were compared with traditional cosmological techniques. The proposed method outperformed the traditional techniques. Gradient-boosted trees were used in [10] to model dark matter halo formation. In [11], the Bayesian optimization for likelihood-free inference (BOLFI) algorithm was used to reconstruct the two-dimensional position and to determine the size of the interaction charge signal. The results showed that the BOLFI algorithm provides improved accuracy of 15% in reconstruction when compared in the case of events at large radii (R > 30 cm, the outer 37% of the detector). The investigation also showed that the proposed algorithm provided smaller uncertainties compared to other methods.

### 1.2. Definition of Novelty and the Research Hypotheses

From the previous short literature overview, complex ML models such as DNN have been used in CDMS detectors' investigation. Although the estimation results were high, the problem is that these algorithms require substantial computational resources. Another problem is that these types of ML methods cannot be easily expressed as an equation.

The authors present a novel approach through the use of genetic programming-symbolic regression (GPSR), which was applied to determine equations that can accurately reconstruct the particle interaction locations in the SuperCDMS. A public dataset [12] provided by a team from the University of Minnesota was used in the research. In comparison to the reviewed research, the research questions can be posed as follows:

- Can an SE be obtained using GPSR that can accurately reconstruct the locations of the interactions in SuperCDMS detectors?
- Is it possible to obtain a set of robust SEs using GPSR with randomly tested hyperparameters and validated through k-fold cross-validation that can reconstruct the locations of the interactions in the SuperCDMS with high accuracy?
- Is it possible to achieve even higher estimation accuracy in the reconstruction of the locations' interactions by combining multiple SEs that were obtained from different GPSR executions?
- Are all input variables required as model inputs to accurately reconstruct the interaction locations?

The presented paper consists of the following sections, i.e., the Materials and Methods, Results, Discussion, and Conclusions. In the Materials and Methods, the detailed dataset description is provided with statistical analysis following the description of the research methodology. In this section, the used method is described. In the Results Section, the best set of SEs obtained after five-fold cross-validation (5-CV) is shown with GPRS hyperparameters used to obtain them. Within the Discussion, the obtained models and results of the statistical analysis are further presented and discussed. Based on the hypotheses, presented results, and discussion, the Conclusions are given in the final section. Besides that, the Conclusions Section provides the pros and cons of the proposed method and possible directions for future work.

## 2. Materials and Methods

The materials, namely the dataset, are given a short description with a basic statistical analysis. Based on the dataset description, a research methodology is provided in which detailed steps are given. Then, the GPSR algorithm is described, as well as the process of developing a random hyperparameter search (RHS) method and the process of obtaining the SEs through the use of the 5-CV. Finally, the computational resources used in this research are described.

### 2.1. Dataset Description

For this investigation, the publicly available dataset from Kaggle [12] was used. The dataset was provided by the team from the University of Minnesota, and their research [13] was also focused on addressing the problem of accurately reconstructing the locations of interactions in the SuperCDMS detectors using machine learning methods.

In [13], the prototype SuperCDMS germanium detector was tested with a radioactive source positioned on a movable stage that can perform scanning from the center of the detector up to the near edge. The SuperCDMS germanium detector is a disk-shaped object that is 10 cm in diameter and 3 cm in height, with phonon sensors placed on the top and bottom surfaces to detect particles from a radioactive source. The sensors were used to measure phonons, i.e., quantized vibrations of the crystal lattice, which are produced from interacting particles and travel from the interaction location to the sensors. The number of phonons and relative arrival time to the particular sensors depends on the interaction and the sensor positions. The output of each sensor channel is a waveform for every interaction. In this experiment, the sensors were grouped into six regions labeled A, B, C, D, E, and F on both sides of the detector, as shown in Figure 1.

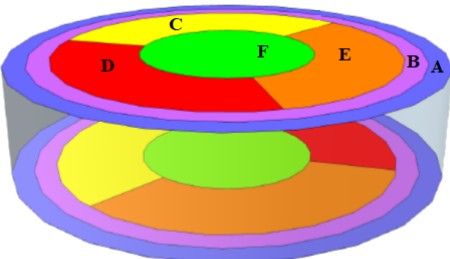

**Figure 1.** The detector regions of the SuperCDMS detector.

To produce interactions at 13 different locations on the detector along a radial path from the center up to the detector's outer edge, a movable radioactive source was used. The 13 different locations are shown in Figure 2, while the numeric values of the locations are listed in Table 1.

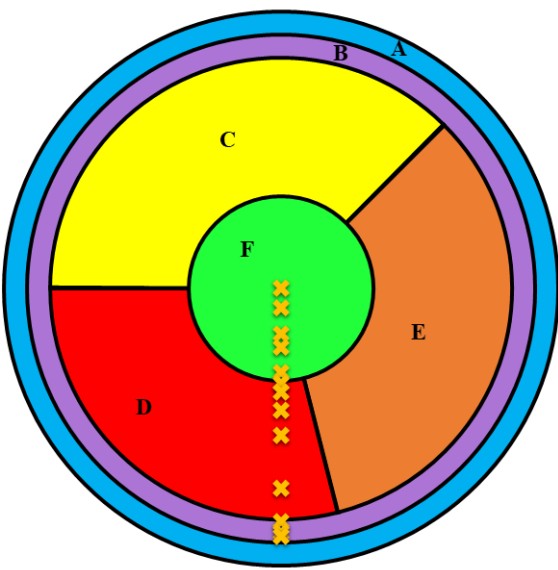

**Figure 2.** Top view showing detector regions (A, B, C, D, E, and F) and interaction locations indicated with the "x" symbol in orange color.

**Table 1.** The measured coordinates of the interaction locations shown in Figure 2.

| Location | 1 | 2 | 3 | 4 | 5 | 6 | 7 | 8 | 9 | 10 | 11 | 12 | 13 |
|---|---|---|---|---|---|---|---|---|---|---|---|---|---|
| Coordinate x | 0 | 0 | 0 | 0 | 0 | 0 | 0 | 0 | 0 | 0 | 0 | 0 | 0 |
| Coordinate y | 0 | −3.96 | −9.98 | −12.502 | −17.992 | −19.7 | −21.034 | −24.077 | −29.5 | −36.116 | −39.4 | −41.01 | −41.9 |

The entire dataset consisted of 21 parameters and 7151 samples. However, the first parameter row number ("Row") was omitted from this research. The dataset was obtained by extracting a set of parameters for each interaction from the signals of five sensors. The extracted parameters are sensitive to interaction location, relative timing between pulses in different channels, and features of the pulse shape. The relative amplitudes are also relevant, but were not included in the dataset due to amplification instabilities in the experiment. For each interaction, the following parameters were recorded:

- B, C, D, and F start—the time at which the signal pulse rises to 20% of the signal peak to Channel A. The variables in the original dataset are labeled as PBstart, PCstart, PDstart, and PDstart, respectively.
- A, B, C, D, and F rise—the time required for the signal to rise from 20 % to 80% of its peak. These variables in the original dataset are labeled as PArise, PBrise, PCrise, PDrise, and PFrise,
- A, B, C, D, and F width—the width of the pulse at 80% of the pulse height. The height was measured in seconds. These variables in the original dataset are labeled as PAwidth, PBwidth, PCwidth, PDwidth, and PFwidth.
- A, B, C, D, and F fall—the time required for a pulse to fall from 40% to 20 % of its peak. These variables in the original dataset are labeled as PAfall, PBfall, PCfall, PDfall, and PFfall.

As seen from the minimum and maximum values shown in Table 2, all the input variables labeled $X_0$ to $X_{18}$ are all very small values in the $10^{-6}$ to $10^{-4}$ range. The target (output) variable, which is $y$, is in the $-41.9$ to $0$ range.

**Table 2.** The results of the statistical investigation of dataset variables including the GSPR variable representation.

| Dataset Variable | Data Points | Mean Value | $\sigma$ | Minimum Value | Maximum Value | GPSR Variable Symbol |
|---|---|---|---|---|---|---|
| PBstart | | $-2.484 \times 10^{-6}$ | $5.346 \times 10^{-6}$ | $-1.800 \times 10^{-5}$ | $1.613 \times 10^{-5}$ | $X_0$ |
| PCstart | | $2.491 \times 10^{-5}$ | $2.596 \times 10^{-5}$ | $-2.160 \times 10^{-5}$ | $6.359 \times 10^{-5}$ | $X_1$ |
| PDstart | | $-1.316 \times 10^{-5}$ | $1.014 \times 10^{-5}$ | $-2.990 \times 10^{-5}$ | $5.149 \times 10^{-5}$ | $X_2$ |
| PFstart | | $-5.240 \times 10^{-6}$ | $2.580 \times 10^{-5}$ | $-3.520 \times 10^{-5}$ | $4.211 \times 10^{-5}$ | $X_3$ |
| PArise | | $1.710 \times 10^{-5}$ | $4.501 \times 10^{-6}$ | $8.464 \times 10^{-6}$ | $2.682 \times 10^{-5}$ | $X_4$ |
| PBrise | | $1.973 \times 10^{-5}$ | $3.843 \times 10^{-6}$ | $9.338 \times 10^{-6}$ | $3.221 \times 10^{-5}$ | $X_5$ |
| PCrise | | $3.012 \times 10^{-5}$ | $7.718 \times 10^{-6}$ | $1.157 \times 10^{-5}$ | $5.354 \times 10^{-5}$ | $X_6$ |
| PDrise | | $1.298 \times 10^{-5}$ | $2.078 \times 10^{-6}$ | $8.705 \times 10^{-6}$ | $3.444 \times 10^{-5}$ | $X_7$ |
| PFrise | | $1.675 \times 10^{-5}$ | $7.026 \times 10^{-6}$ | $8.060 \times 10^{-6}$ | $3.158 \times 10^{-5}$ | $X_8$ |
| PAfall | 7151 | $2.148 \times 10^{-5}$ | $1.826 \times 10^{-5}$ | $1.521 \times 10^{-5}$ | $6.814 \times 10^{-5}$ | $X_9$ |
| PBfall | | $2.516 \times 10^{-5}$ | $3.781 \times 10^{-5}$ | $1.596 \times 10^{-5}$ | $9.416 \times 10^{-5}$ | $X_{10}$ |
| PCfall | | $2.712 \times 10^{-5}$ | $4.878 \times 10^{-5}$ | $1.388 \times 10^{-5}$ | $9.884 \times 10^{-5}$ | $X_{11}$ |
| PDfall | | $2.588 \times 10^{-5}$ | $4.621 \times 10^{-5}$ | $1.415 \times 10^{-5}$ | $9.806 \times 10^{-5}$ | $X_{12}$ |
| PFfall | | $2.472 \times 10^{-5}$ | $5.361 \times 10^{-5}$ | $1.163 \times 10^{-5}$ | $9.355 \times 10^{-5}$ | $X_{13}$ |
| PAwidth | | $1.696 \times 10^{-5}$ | $3.659 \times 10^{-5}$ | $7.105 \times 10^{-5}$ | $2.271 \times 10^{-5}$ | $X_{14}$ |
| PBwidth | | $2.071 \times 10^{-5}$ | $3.343 \times 10^{-5}$ | $7.133 \times 10^{-5}$ | $2.983 \times 10^{-5}$ | $X_{15}$ |
| PCwidth | | $2.467 \times 10^{-5}$ | $2.453 \times 10^{-5}$ | $1.648 \times 10^{-5}$ | $3.443 \times 10^{-5}$ | $X_{16}$ |
| PDwidth | | $1.494 \times 10^{-5}$ | $5.623 \times 10^{-5}$ | $3.603 \times 10^{-5}$ | $3.033 \times 10^{-5}$ | $X_{17}$ |
| PFwidth | | $1.765 \times 10^{-5}$ | $7.636 \times 10^{-5}$ | $3.321 \times 10^{-5}$ | $2.665 \times 10^{-5}$ | $X_{18}$ |
| y | | $-21.758$ | $14.091$ | $-41.9$ | $0$ | $y$ |

The extremely low values of the input variables in the GPSR can lead to poor performance, i.e., the low estimation accuracy of the obtained SEs. To improve the performance of GPSR, the StandardScaler technique [14,15] was applied to the input dataset variables. The standard scaling technique is a technique that standardizes dataset variables and scales them to the unit variance. The standard manner of scaling the variable values is calculated per:

$$z = \frac{x - \sigma}{s}, \tag{1}$$

with $m$ and $s$ being the mean and $\sigma$ of the dataset variable. The idea is to apply the StandardScaler method to all input variables before using the dataset in the GPSR algorithm.

Besides the statistical analysis in the dataset analysis, it is important to perform the correlation analysis. In this paper, Pearson's correlation analysis [16] was performed to determine the correlations between the output and individual inputs. The obtained correlation value between two dataset variables is calculated in the range of $<-1.0, 1.0>$. The absolute value of the correlation result is proportional to the rate at which two variables change together—with a positive value indicating a mutual increase and a negative value indicating that one variable decreases while the other increases. A heat map of the calculated coefficients is given in Figure 3.

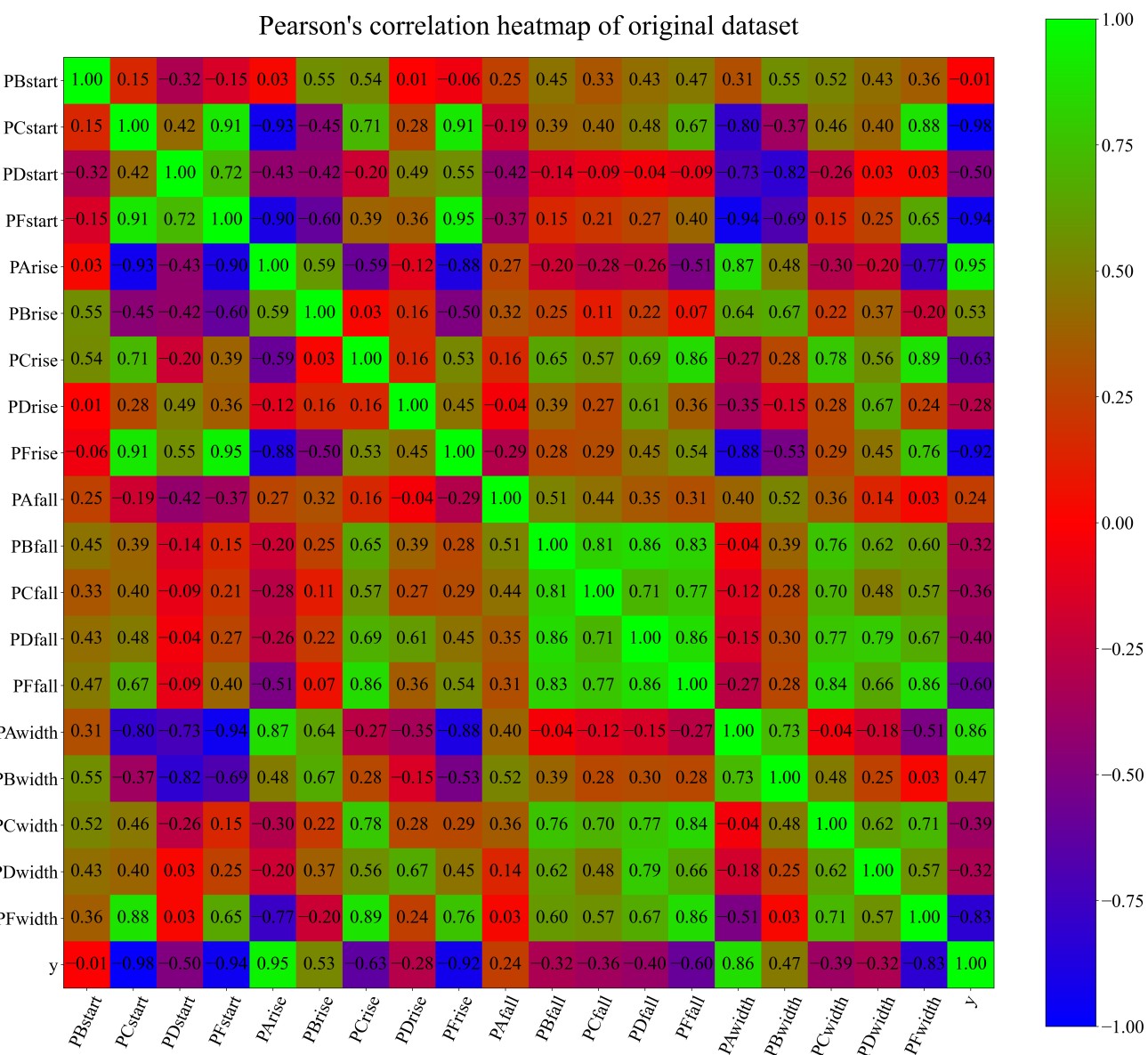

**Figure 3.** Pearson's correlation heat map.

As seen from Figure 3, the highest negative correlation (−0.98–−0.83) was achieved between PCStart, PFStart, PFrise, PFwidth, and the output variable y. The highest positive correlation (0.86–0.95) was achieved between PAwidth, PArise, and output variable y, respectively. The poorest correlation coefficient was achieved in the cases of PBstart-y (−0.01), PBfall-y (−0.32), PCfall-y (−0.36), PCwidth-y (−0.39), and PCwidth-y (−0.32). From the correlation heat map, it can be noticed that PBFall, PCfall, PDfall, and PFfall are mutually highly correlated variables, while showing a poor correlation in regard to the output. However, since GPSR is not a computationally intensive algorithm, it is good practice to include all input variables in the investigation. The reason why all input variables were included was to compare the variables used in the final models with their correlations.

### 2.2. Research Methodology

Figure 4 demonstrates the flow of the performed research methodology.

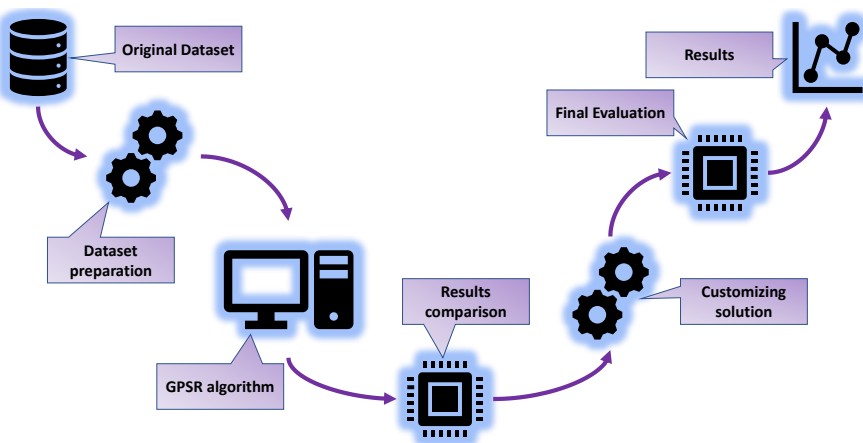

**Figure 4.** The flow chart of the research methodology.

Based on the previous flowchart, the research methodology can be summarized in the following steps:

- Dataset preparation—Checking the dataset for null values and deleting the first column ("Row"), which is not relevant for the analysis. Separating the dataset into input and output variables and applying of StandardScaler method on the input dataset variables. Dividing the dataset into training and testing datasets in a 70:30 ratio.
- GPSR algorithm—The GPSR algorithm is combined with a RHS to find the hyperparameters that yield the best-performing models. Perform training of GPSR on the training dataset using cross-validation, for the evaluation of the testing dataset.
- Result comparison—Perform a comparison of the best sets of SEs in terms of the estimation accuracy.
- Customizing solution—Combining the five best SEs to obtain a robust estimator for the reconstruction of the interaction locations.
- Final evaluation—Perform the final evaluation of the customized solution on the entire dataset.

### 2.3. Genetic Programming-Symbolic Regression

Genetic programming-symbolic regression (GPSR) begins its execution by creating the unit population. The population members' quality initially is very low. Then, through a consecutive number of generations, they are fit for a specific task with the application of genetic operations.

To build the initial population in GPSR, the dataset is required to have labeled input variables, the target output variable, the range of constant values, and mathematical functions. It should be noted that, in this paper, the following mathematical functions were used: addition, subtraction, multiplication, division, square root, cube root, absolute value, sine, cosine, tangent, minimum value, maximum value, natural logarithm, logarithm with base 2 and 10, respectively. It should be noted that mathematical functions such as division, natural logarithm, logarithm with base 2 and 10, and square root are specifically defined to avoid zero division errors and infinite or complex values during GPSR execution. The definition of these functions is given in Appendix A.The constant range hyperparameter is defined in Table 3. From these three sets, the GPSR randomly selects components to create population members, i.e., initial population. The hyperparameters required for the development of the initial population are population size, the maximum number of generations, and the constant range.

According to [17], there are three commonly used methods for creating population members, and these are full, grow, and ramped-half-and-half. The ramped-half-and-half

method is a combination of the full and grow methods, i.e., both methods are used to create half of the dataset. To ensure a higher population diversity in the ramped-half-and-half method, the population members' depth is set in the range of hyperparameter init_depth. For example, if the init_depth value is set to (6,20), this means that the population members will be created with the full and grow method, where the population member's depth will be in the range from 6 to a maximum of 20.

The population members (SEs) and obtained SEs from each GPSR execution can be measured in terms of length. The length is counted as the sum of all mathematical functions, variables, and constants contained in a single SE. In the Results Section, besides the depth of the SEs, the length of the SEs will be also given.

In this paper, the fitness function used in all investigations is the mean absolute error (*MAE*), which can be written in the following form:

$$MAE = \frac{\sum_{i=1}^{n}(y_i - x_i)}{n},\qquad(2)$$

where $y_i$, $x_i$, and $n$ are predicted values by the population member, the true (output) dataset value, and the number of dataset samples [18]. To calculate the *MAE*, first, the population member must be evaluated, i.e., the values of the input variables must be provided to calculate the output. This output is the predicted $y_i$ variable in Equation (2).

The tournament selection method was used to select the parents to which the genetic operations will be applied. When this selection is used, the population members are selected randomly for comparison. The member with the best fitness value is chosen as the winner of the process. The tournament_size hyperparameter defines the number of units used in this process.

Sometimes, during the execution of GPSR and due to the low correlation between dataset variables, the size of candidate solutions can rapidly grow through the generations. The size of the population members can grow so large that this can result in the bloat phenomenon. Various methods can be used to prevent this phenomenon, such as size fair crossover [19], size fair mutation [20], the Tarpeian method [21], and the parsimony pressure method [22]. The latter method is the most-commonly used and will be used in this paper. This method penalizes large programs during tournament selections by increasing the fitness value score, so they are not selected as the winners of tournament selection. This hyperparameter used for preventing the bloat phenomenon is parsimony_coefficient, and it is one of the most-sensitive coefficients to adjust, so when defining its range, extensive initial testing is required.

The execution of GPSR like in the majority of evolutionary algorithms can go indefinitely if the termination criteria are not defined. In GPSR, this is achieved with the stopping_criteria hyperparameter and the generations hyperparameter. The stopping_criteria is the fitness function value, which will stop the execution if achieved. If this fitness is not achieved, the hyperparameter generations will stop the execution when the number of iterations is equal to it. In all the performed investigations, the execution was stopped due to the generations parameter, due to the small value of the stopping_criteria. After each tournament selection, the best population member is obtained, and in that population member, one of the genetic operations is performed. Four genetic operators were selected in GPSR. The first of them is a crossover, the second a subtree mutation, the third a hoist mutation, and the fourth a point mutation. Each hyperparameter represents the probability and the possible sum of all genetic operations that must be equal to 1 to prevent tournament selection winners from being cloned and entering in the next generation. In this paper, the sum of all genetic operation probabilities was lower than one, so some winners enter the next generation unchanged.

In the case of a crossover operation, two winners are required, which determine the parent and donor. On the parent and the donor, random subtrees are selected, and the subtree from the donor replaces the subtree of the parent to form the member of the next generation. However, in the case of subtree mutation, the selection is a little different.

To select a subtree mutation, only one tournament winner is needed, which is later replaced with a randomly generated subtree created using mathematical functions, the constants from a constant range, and the input variables from the dataset. A similar situation occurs with the hoist mutation, which also requires only one tournament winner on which the random tree is defined. Then, a random node is selected on that subtree, which replaces the entire subtree. The point mutation demands only one tournament winner, then afterward, random nodes are selected. The constants are then replaced by randomly selected constants from the constant range, and also, the variables are replaced by the randomly selected input variables and the other functions by other randomly selected functions. However, in the case of the functions, the arity of randomly selected functions must be the same as the function in the tournament winner.

### 2.4. Random Hyperparameter Search

To find the optimal combination of hyperparameters, the RHS method was used. To develop this method for GPSR, the initial testing of each hyperparameter was required.

The GPSR hyperparameter ranges defined after initial testing are listed in Table 3.

**Table 3.** The range of GPSR hyperparameters.

| Hyperparameter Name | Range |
|---|---|
| population_size | 1000–2000 |
| number of generations | 100–200 |
| tournament_selection | 10–500 |
| init_depth | 3–15 |
| crossover | 0.001–1.0 |
| subtree_mutation | 0.001–1.0 |
| hoist_mutation | 0.001–1.0 |
| point_mutation | 0.001–1.0 |
| stopping_criteria | $0–1 \times 10^{-8}$ |
| maximum_samples | 0.99–1 |
| constant_range | −10,000–10,000 |
| parsimony_coefficient | $0–1 \times 10^{-4}$ |

In the case of GPSR, there were four parameters that were most-influential: the size of the population (population_size), the number of possible generations (number_of_generations), the tree depth (init_depth), and finally, the parsimony coefficient [23]. The population size and the number of possible generations are highly correlated hyperparameters. If the size of the population and number of generations is too large, it can lead to a very long execution time. The ranges shown in Table 3 proved to be optimal. Initially, init_depth was increased, but the GPSR execution showed that higher values of the init_depth hyperparameter can lead to longer execution times without any benefit to the estimation accuracy of the obtained SE. As the stated parsimony_coefficient is the most-crucial and most-sensitive hyperparameter to define, the range is defined by trial and error, so the range shown in Table 3 will prevent the occurrence of bloat phenomena, but will allow the stable growth of the population members.

The initial assumption was that crossover or subtree mutation will be the most-influential genetic operation in the evolution process. However, an initial investigation showed that very high values of these two genetic operations can lead to local minimums, i.e., the fitness function value is constant, while the size of the population members grows rapidly. Therefore, the value of all four genetic operations was set to the 0.001–1 range.

In this way, the sum of all possible genetic operations was nowhere near one, so some tournament selection winners entered the next generation unchanged.

It should be noted that each GPSR execution is terminated after reaching the defined number of maximum generations. As described, the GPSR algorithm has two termination criteria, i.e., the maximum number of generations and the stopping criteria. However, the stopping criterion, i.e., the value of the fitness function, was set to an extremely small value to ensure that the execution of the GPSR algorithm is terminated after reaching the maximum number of generations.

The maximum samples were set in the 0.99 to 1 range, so each time the fitness function was computed, almost the entire training dataset was used. If the value of the maximum samples was set to 1, then during the GPSR execution, the out-of-bag raw (OOB) fitness would not be shown. OOB fitness refers to the fitness of each population member in held-out samples. To see OOB fitness during execution, the maximum samples must be less than 1. To ensure that the population's growth is stable, a relatively large range of constants was provided to ensure that the small range of constants can lead to the large growth of members.

### 2.5. Training and Evaluation Process with the GPSR Algorithm

The entire flowchart of the training process with GPSR is shown in Figure 5.

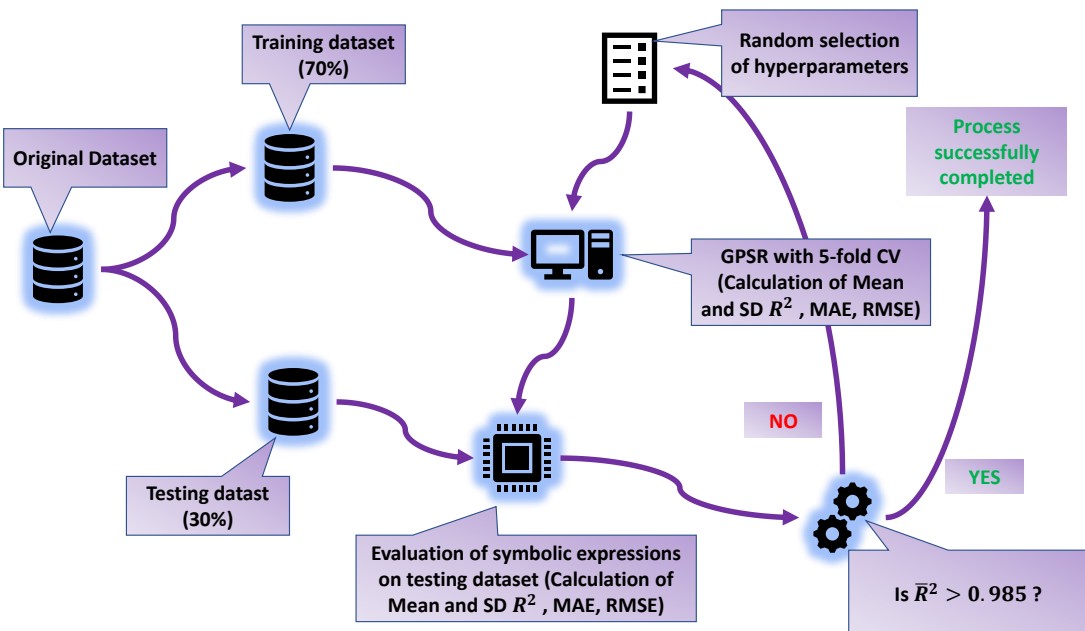

**Figure 5.** The flowchart of obtaining symbolic expressions using the GPSR algorithm.

Due to a large number of samples (7157), the original dataset after preprocessing with StandardScaler (only input variables) was divided into a 70:30 ratio: 70% of the dataset was used in the 5-CV and the remaining 30% for the final evaluation.

The investigated process begins with randomly selecting GPSR hyperparameters (range values defined in Table 3), then the execution of GPSR algorithm with the 5-CV on the training dataset. In the 5-CV [24], there are 5 splits, i.e., the GPSR execution is trained 5 times, and for each time, a new SE is obtained. After each split, the performance metrics are calculated on the training and validation folds. After the process of training is performed, the mean and $\sigma$ values are calculated and stored. When this step is performed, the obtained SEs are evaluated on the testing dataset, and the performance metrics (mean and $\sigma$) values are obtained. For the termination criteria of the entire process, the mean value of $R^2$ must be greater than 0.985. If the value is above 0.985, the process is terminated; otherwise, the process begins from the beginning with a random selection of the GPSR hyperparameters.

### 2.6. Performance Metrics and Methodology

In this paper, to evaluate the SEs, three performance metrics were used, i.e., the coefficient of determination ($R^2$) [25], the mean absolute error $MAE$ [18] (Equation (2)), and the root-mean-squared error ($RMSE$) [26]. The $R^2$ is in the 0 to 1 range, where 0 represents the worst-possible value, while the value of 1 represents the best-possible value and is aimed for. Regarding the $MAE$ and $RMSE$ values, the goal is to obtain as low a value as possible.

As stated in the description of the GPSR algorithm, the $MAE$ metric was used to evaluate the population members (fitness function). After each SE is obtained, it is evaluated to determine the $R^2$, $MAE$, and $RMSE$ values. To calculate the mean and $\sigma$ values of the performance metrics in GPSR with RHS and the 5-CV, the following steps are performed:

- During the 5-CV, calculate the performance metric values on the training and validation sets.
- After the 5-CV process is complete, calculate the performance metrics (mean values obtained on the training dataset).
- Perform the last evaluation of the trained model (five SEs) on the testing dataset, and calculate the performance (evaluation) metric values (values achieved on the testing dataset),
- Calculate the mean and $\sigma$ values of the performance metrics from the obtained values on the training and testing datasets.

### 2.7. Computational Resources

The entire research was performed on a desktop computer consisting of an Intel I7-4770 processor supported by 16 GB of DDR3 RAM. All scripts were written and made in the Python Programming language (Version 3.9.12). The statistical analysis was conducted using pandas [27] and the matplotlib [28] library. The datasets were scaled using the StandardScaler method from the scikit-learn library (Version 1.2.0) [14]. GPSR was used from the gplearn library (Version 0.4.1.) [29].

## 3. Results

The Results Section is divided into two subsections entitled "Results obtained using GPSR algorithm with RHS and 5-CV" and "Custom solution and final evaluation". In Section 3.1, the estimation accuracy of the three best cases of the SEs is presented, obtained during the 5-CV (training dataset) and final evaluation (test dataset). In Section 3.2, the custom set of the five SEs was created by picking the SEs with the highest estimation accuracy from the previously presented subsection. Finally, the performance evaluation of the modified (customized) set of SEs on the entire dataset is presented.

### 3.1. Results Acquired Using GPSR with RHS and 5-CV

The GPSR algorithm with the 5-CV was executed multiple times. Before each execution, the hyperparameters were randomly selected from the predefined ranges shown in Table 3. From the obtained results, the three best cases were selected that achieved the highest estimation accuracy. The combination of the hyperparameters that were used to obtain the highest estimation accuracies is listed in Table 4.

From Table 4, it can be seen that, for Cases 1 and 3, the population size was very large. In Case 1, the point mutation (0.43) was the dominating genetic operation, while in Cases 2 and 3, the crossover (0.41) and subtree mutation (0.469) dominated the genetic operations. However, in Case 3, both crossover and subtree mutation had higher probabilities than the other two types of mutations. The stopping criteria, as planned, were never reached by any population member in all three cases, and each execution stopped after the maximum possible number of generations was attained. The parsimony coefficient (Pcoef) value was much higher in Cases 2 and 3 than in Case 1. The results of these three cases are shown in Figure 6 and Table 5.

**Table 4.** The combination of the GPSR hyperparameter values for which the highest estimation accuracy was achieved.

| Case No. | GPSR Hyperparameters (Population Size, Number of Generations, Tournament Selection, Init_Depth, Crossover, Subtree_Mutation, Hoist_Mutation, Point_Mutation, Stopping_Criteria, Max_Samples, Constant_Range, Parsimony_Coefficient) |
|---|---|
| 1 | 1978, 233, 254, (6, 15), 0.043, 0.22, 0.22, 0.43, $5.49 \times 10^{-7}$, 0.99, $(-9793.08, 1402.72)$, $9.15 \times 10^{-6}$ |
| 2 | 1075, 196, 461, (3, 15), 0.41, 0.4, 0.05, 0.097, $9.95 \times 10^{-7}$, 0.99, $(-8821.29, 3713.89)$, $9.2 \times 10^{-4}$ |
| 3 | 1500, 171, 108, (7, 11), 0.45, 0.469, 0.043, 0.0071, $8.17 \times 10^{-7}$, 0.99, $(-4076.61, 4272.87)$, $7.16 \times 10^{-4}$ |

The Splits 1–5 shown in Figure 6 and Table 5 indicate the GPSR algorithm execution on the different training sets (4 folds) and the evaluation on the validation set (1 fold) in the 5-CV. After each split, the performance metric values were calculated on the training and validation set and the mean and $\sigma$ values were calculated. When all five splits were performed, the total mean and $\sigma$ of all assessment (evaluation) metrics were calculated. These "Total" values represent the final values of the 5-CV process. It should be noted that, after each split, the SE was obtained, so each case consisted of five SEs.

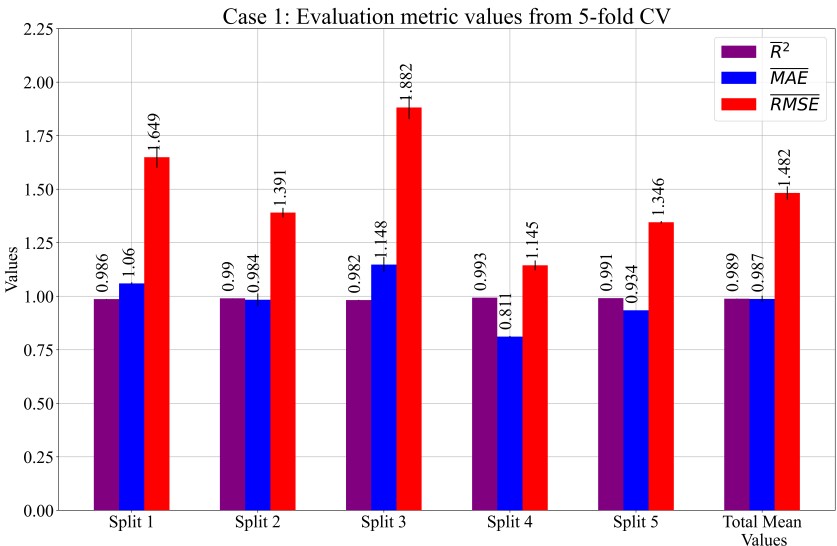

**Figure 6.** *Cont.*

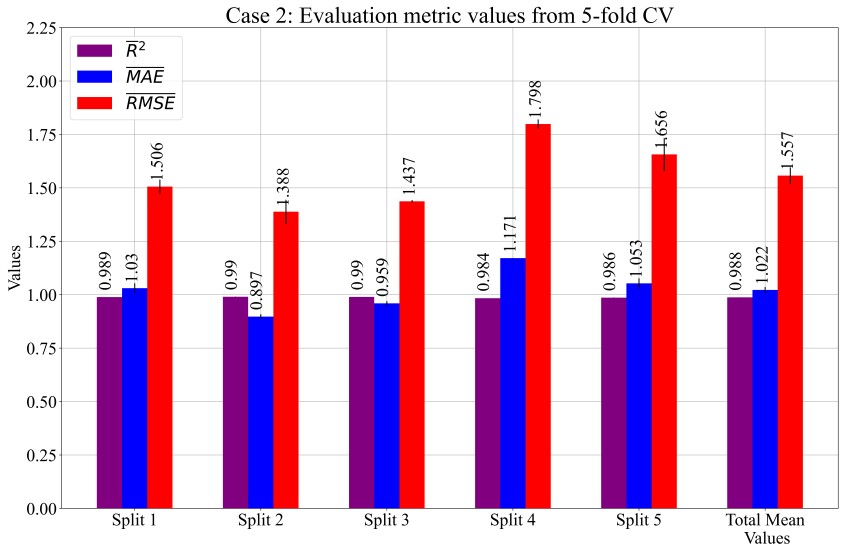

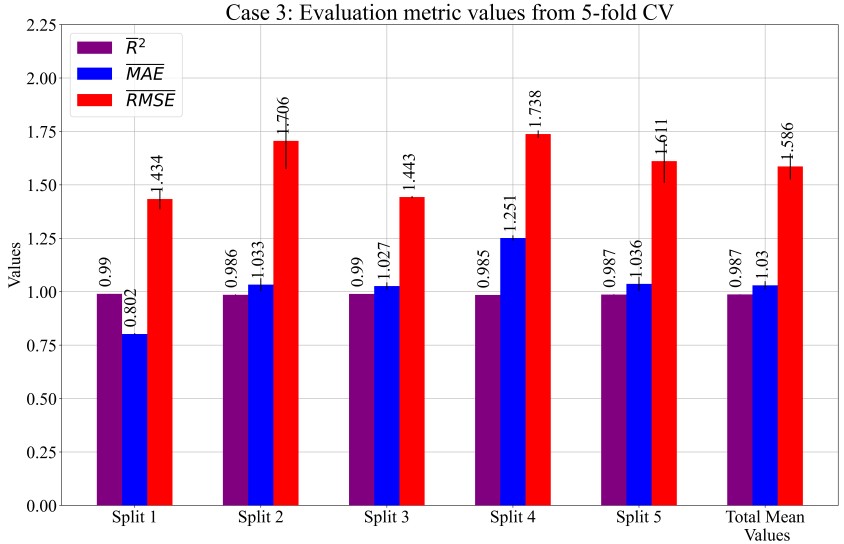

**Figure 6.** The mean values of the obtained performance metric during GPSR 5-CV. The $\sigma$ values are presented as error bars.

As seen from Figure 6 and Table 5, in all three cases, the mean performance metric values obtained with the 5-CV on the training data were very high with low $\sigma$ values. Table 6 shows the depth and length of each SE with the average depth and length of the SEs in each case.

To clarify the terms length and depth in Table 6, the length is a representation of several elements (constants, functions, and possible variables) in the SEs, while the depth is measured when the SE is shown in tree form. The depth is measured from the required root node to the deepest leaf of the SE. Since the tree form of the SE is not important for this investigation, the length of the SE will be investigated. The highest SE lengths and average length were achieved in Case 1 followed by Case 3 and Case 2. From the length and the estimation accuracy in Table 5, the best case is Case 2, since the estimation accuracy is slightly lower than Case 1, while the average length of the SEs is the lowest.

**Table 5.** The mean and $\sigma$ values of $R^2$, $MAE$, and $RMSE$ were achieved in a 5-fold cross-validation process.

| | Performance metric | Split 1 | Split 2 | Split 3 | Split 4 | Split 5 | Total |
|---|---|---|---|---|---|---|---|
| **Case 1** | $\overline{R}^2$ | 0.9864 | 0.9903 | 0.9821 | 0.9935 | 0.9909 | 0.9886 |
| | $\sigma(R^2)$ | 0.0009 | 0.0003 | 0.0013 | 0.0002 | 0.0001 | 0.0006 |
| | $\overline{MAE}$ | 1.0599 | 0.9837 | 1.1479 | 0.8113 | 0.9342 | 0.9874 |
| | $\sigma(MAE)$ | 0.0057 | 0.0291 | 0.0354 | 0.0035 | 0.0003 | 0.0148 |
| | $\overline{RMSE}$ | 1.6494 | 1.3905 | 1.8818 | 1.1445 | 1.3456 | 1.4824 |
| | $\sigma(RMSE)$ | 0.0501 | 0.0222 | 0.0542 | 0.0230 | 0.0057 | 0.0311 |
| | Performance metric | Split 1 | Split 2 | Split 3 | Split 4 | Split 5 | Total |
| **Case 2** | $\overline{R}^2$ | 0.9889 | 0.9905 | 0.9896 | 0.9835 | 0.9860 | 0.9877 |
| | $\sigma(R^2)$ | 0.0001 | 0.0006 | 0.0001 | 0.0001 | 0.0009 | 0.0004 |
| | $\overline{MAE}$ | 1.0304 | 0.8973 | 0.9593 | 1.1713 | 1.0530 | 1.0223 |
| | $\sigma(MAE)$ | 0.0233 | 0.0115 | 0.0108 | 0.0006 | 0.0238 | 0.0140 |
| | $\overline{RMSE}$ | 1.5057 | 1.3883 | 1.4369 | 1.7984 | 1.6564 | 1.5571 |
| | $\sigma(RMSE)$ | 0.0331 | 0.0562 | 0.0062 | 0.0218 | 0.0772 | 0.0389 |
| | Performance metric | Split 1 | Split 2 | Split 3 | Split 4 | Split 5 | Total |
| **Case 3** | $\overline{R}^2$ | 0.9899 | 0.9856 | 0.9896 | 0.9846 | 0.9868 | 0.9873 |
| | $\sigma(R^2)$ | 0.0004 | 0.0019 | 0.0001 | 0.0001 | 0.0013 | 0.0008 |
| | $\overline{MAE}$ | 0.8020 | 1.0330 | 1.0266 | 1.2512 | 1.0365 | 1.0299 |
| | $\sigma(MAE)$ | 0.0041 | 0.0302 | 0.0170 | 0.0128 | 0.0328 | 0.0194 |
| | $\overline{RMSE}$ | 1.4336 | 1.7058 | 1.4429 | 1.7375 | 1.6111 | 1.5862 |
| | $\sigma(RMSE)$ | 0.0489 | 0.1311 | 0.0051 | 0.0170 | 0.1016 | 0.0608 |

The performance metric mean and $\sigma$ values obtained for each case on the testing dataset are shown in Figure 7 and Table 7.

**Table 6.** The symbolic expressions length and depth of each case with average length and depth.

| Case No. | Symbolic Expression No. | Length | Depth | Average Length | Average Depth |
|---|---|---|---|---|---|
| 1 | 1 | 220 | 33 | 206.6 | 28.4 |
| | 2 | 257 | 41 | | |
| | 3 | 106 | 13 | | |
| | 4 | 351 | 36 | | |
| | 5 | 99 | 19 | | |

**Table 6.** *Cont.*

| Case No. | Symbolic Expression No. | Length | Depth | Average Length | Average Depth |
|---|---|---|---|---|---|
| 2 | 1 | 75 | 17 | 81.4 | 16.2 |
| | 2 | 87 | 16 | | |
| | 3 | 109 | 18 | | |
| | 4 | 58 | 15 | | |
| | 5 | 78 | 15 | | |
| 3 | 1 | 145 | 31 | 161.6 | 24.8 |
| | 2 | 156 | 19 | | |
| | 3 | 127 | 27 | | |
| | 4 | 230 | 17 | | |
| | 5 | 150 | 30 | | |

**Table 7.** The performance metric values obtained on the test dataset.

| Case Number | Evaluation Metric | Symbolic Expressions | | | | | Mean | $\sigma$ |
|---|---|---|---|---|---|---|---|---|
| | | 1 | 2 | 3 | 4 | 5 | | |
| 1 | $R^2$ | 0.9827 | 0.9892 | 0.9811 | 0.9934 | 0.9905 | 0.9876 | 0.0048 |
| | $MAE$ | 1.1180 | 0.9919 | 1.1645 | 0.8225 | 0.9619 | 1.0096 | 0.1215 |
| | $RMSE$ | 1.8649 | 1.4646 | 1.9488 | 1.1537 | 1.3810 | 1.5506 | 0.3050 |
| 2 | $R^2$ | 0.9878 | 0.9896 | 0.9887 | 0.9836 | 0.9837 | 0.9867 | 0.0025 |
| | $MAE$ | 1.0615 | 0.9393 | 0.9894 | 1.1815 | 1.1450 | 1.0634 | 0.0911 |
| | $RMSE$ | 1.5664 | 1.4490 | 1.5057 | 1.8173 | 1.8116 | 1.6300 | 0.1551 |
| 3 | $R^2$ | 0.9909 | 0.9865 | 0.9889 | 0.9828 | 0.9873 | 0.9873 | 0.0027 |
| | $MAE$ | 0.8032 | 1.0518 | 1.0513 | 1.3162 | 1.0666 | 1.0578 | 0.1623 |
| | $RMSE$ | 1.3529 | 1.6464 | 1.4958 | 1.8588 | 1.5978 | 1.5903 | 0.1677 |

As seen from Figure 7 and Table 7, the mean values of the performance metric are almost the same as those obtained on the training dataset. The $\sigma$ values are slightly higher when compared to those obtained on the training dataset. Although Case 2 showed better performance on the training dataset, the results of the testing dataset showed that Case 3 slightly outperformed Case 2.

It can be noticed that the highest scores in Table 7 are marked with red color. The marked scores represent the chosen SE of three cases that will be used for a customized solution.

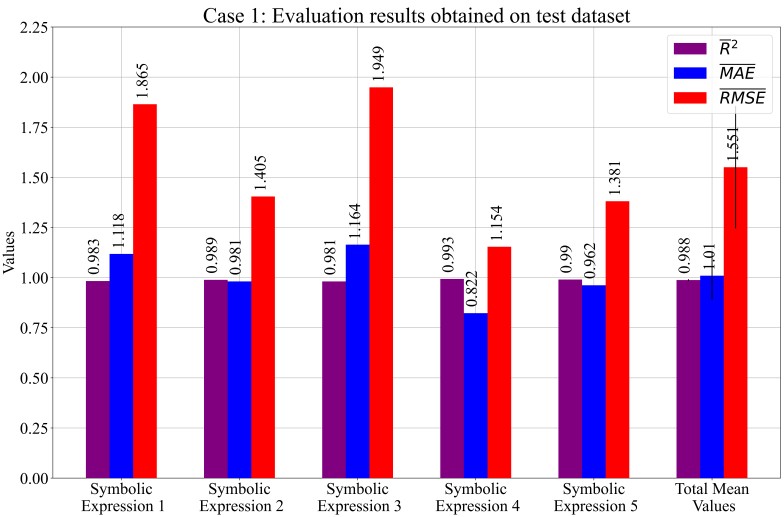

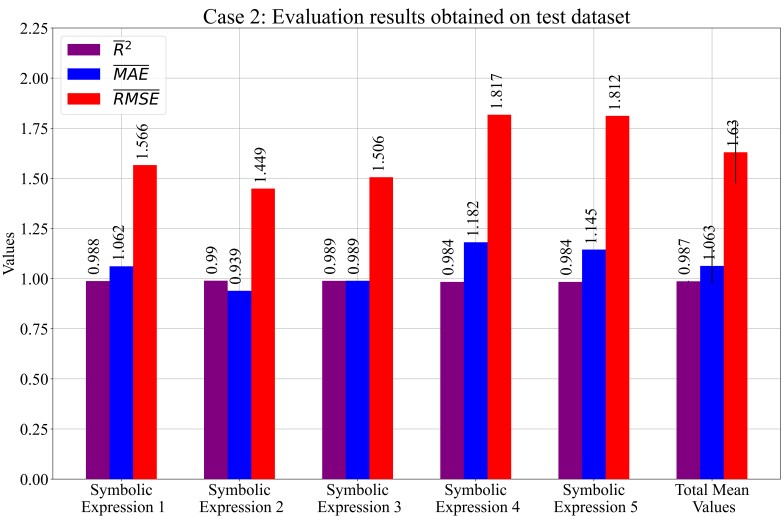

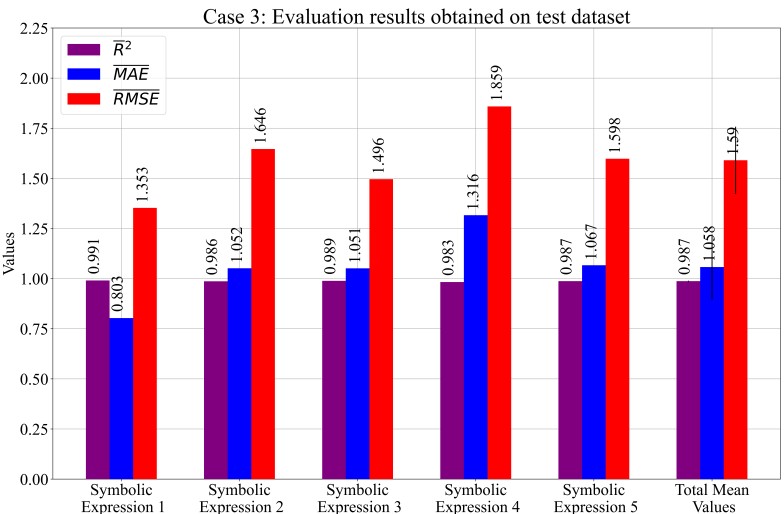

**Figure 7.** The mean values of the performance metric obtained on the testing dataset. The $\sigma$ values are presented as error bars.

### 3.2. Combination of the Best SE

Based on the results obtained on the test dataset, the five SEs with the best estimation accuracy were selected. In Table 7, the results of the best SEs that were selected are marked with red color. The main criterion for choosing the SEs was that the $\overline{R}^2$ score was higher than 0.989. The chosen SEs can be written in the following form:

$$
\begin{aligned}
y_1 \;=\; &-14.061\Bigg(\Big|\sin\Big(\sqrt{\sqrt[3]{\frac{\min(\sin(X_1),\,\sqrt[3]{X_0})}{\sqrt{X_9}}}}\Big) - \min(-X_{18}+X_4 - |\min(\sqrt{X_{18}}, -X_{18}-X_3 \\
&+\; X_4 + X_6 - |X_2 + X_5 - \min(-X_3 + X_6 + \cos(\cos(X_5)), -X_{18}-X_3+X_4+X_6 \\
&-\; | - |\max(X_2+X_5, 1.44\log(\tan(X_3))) - \min(X_5 + \max(X_{17},X_{18}), \sqrt[3]{X_{15}}, -X_{18}-X_3 \\
&+\; 2X_4 - |X_7| - 0.43\log(X_1) - 0.43\log(X_2) + \max(X_{17},X_{18}) - \sqrt[3]{X_1} + \sqrt[3]{X_{15}} - \tan(\sqrt{X_4}))| \\
&-\; \cos(\cos(X_1+X_{17})) + \cos(0.43\log(X_1))| - 0.43\log(X_1))| - \cos(\sin(X_8)))| - \cos(\min(X_3,X_6)) \\
&+\; 0.43\log(\sqrt{\cos(\cos(X_5)))}, \sin(\sqrt{\sqrt[3]{\frac{\sin(X_1)}{\sqrt{X_9}}}}))|\Bigg)^{\frac{1}{3}},
\end{aligned}
\tag{3}
$$

$$
\begin{aligned}
y_2 \;=\; &0.43\log(X_5 - 1.44\log(\sin(\max(X_1,X_{14})))) + 0.43\log(X_5 - \sin(\log(\max(X_1,X_{14})))) \\
&+\; 0.43\log(\max(X_5, X_5 - X_6) - \log(\max(X_1,X_5))) - 4X_1 + 4X_{14} + 0.43\log(0.43\log(X_{14})) \\
&+\; X_{15} - 3X_{18} + \sin(\sin(1.44\log(X_{18}))) + \cos(X_2(X_2-X_6)) - X_3 + X_4 + \sin(X_5) - X_6 \\
&+\; 1.129\sqrt[3]{\log(X_6)} + X_8 - 20.4537,
\end{aligned}
\tag{4}
$$

$$
\begin{aligned}
y_3 \;=\; &-\max(X_0X_{14}+X_4, X_2 + 2X_3 + 2X_5, 1.44\log(X_7), \max(X_6, |X_2|)) \\
&+\; \min(X_{18},X_4)\min(X_4,X_6)) - \min(X_{18},X_4) + \min(-3X_3 - 2X_5, 0.43\log(\cos(\min(X_4,X_6))) \\
&+\; 2X_8))\log(\sqrt{X_1}) - X_{16} - X_{18} - 4X_3 + 6X_4 + 2X_5 - \cos(X_6) - 19.2567,
\end{aligned}
\tag{5}
$$

$$
\begin{aligned}
y_4 \;=\; &\min\Big(X_5, \Big(-\max(\sqrt[3]{|X_{17}|-X_1}, \sqrt[3]{\log(X_{12})}) - |X_7 + \log(1.44\log(\max(X_1,X_{17})))| \\
&+\; 1.44\log(\cos(\max(|X_2|, \tan(X_1)))) - \min(|X_5|, \Big(\min(X_4-X_0, 1.44\log(X_{12}(X_{18}-X_4))) \\
&+\; \sqrt[3]{\min(X_0,X_2)} - |\log(X_{17})| - \sin(X_{17})\Big)^{\frac{1}{3}} + \sqrt[3]{\min(X_0,X_2)} - 1.44\log(\sin(|X_3|)) - X_{10} \\
&-\; 1.44\log(\cos(\min(X_2,X_3))) - X_0) - \Big(\sqrt[3]{\min(X_0,X_2)} - 1.44\log(\cos(|X_2|)) \\
&+\; 0.43\log(\log(X_{13})) - \sin(X_{17})\Big)\frac{1}{3} + \sqrt[3]{\min(X_0,X_2)} - 1.44\log(\cos(|X_2|)) \\
&-\; \Big(\sqrt[3]{\sqrt[3]{\min(X_0,X_2)} - 1.44\log(\cos(|X_2|)) + 0.43\log(\log(X_{13})) - \sin(X_{17}) + \sqrt[3]{\min(X_0,X_2)}} \\
&-\; 1.44\log(\cos(\min(X_0,X_2)))\Big)^{\frac{1}{3}} - \Big(\min(X_4-X_0, |X_5|) + \sqrt[3]{\min(X_0,X_2)} - 1.44\log(\cos(|X_2|)) \\
&-\; \sin(X_{17})\Big)^{\frac{1}{3}} - \min(X_5, 1.44\log(X_0) - |X_0|) + \max(\min(X_{12},X_{15}) + X_{13}, X_0X_6 + \sin(X_7)) \\
&-\; \min(0.43\log(X_{16}), \max(-1255.06, X_0, X_{18})) + 1.44\log(\min(X_4-X_0, \max(-1255.06, X_0, \\
&\quad X_{18}))) + 1.44\log(\cos(\max(\sin(\cos(X_{13})), \sqrt[3]{X_{15}} - \sqrt[3]{X_1}))) + 1.44\log(\cos(\max(-9072.8, \\
&\quad \tan(X_1)))) - \tan(\min(X_0, \cos(X_{14}+X_4))) - 3\sqrt[3]{\min(X_0,X_2)} - \min(X_{14},X_6) \\
&+\; 1.44\log(\cos(\min(X_2,X_3))) + 1.44\log(\cos(\min(X_2, \log(X_2)))) - |X_2 + X_7 + \log(X_0)| \\
&-\; |X_7 + X_8 + \log(X_0)| - 1.44\log(\log(X_0) + X_7) - X_0 + 1.44\log(\cos(X_0)) + X_1 + X_{10} - X_{11} \\
&+\; 2X_{12} + \sqrt[3]{X_{12} - 3012.83} - X_{13} + \cos(X_{17}) + \sin(\cos(X_{17})) - X_2 + X_4 + X_7 \\
&-\; 1.44\log(X_7) + X_8 - X_9\Big)^{\frac{1}{3}} + \min(X_0,X_1) + \min(-19.0691, \sin(X_{18}))
\end{aligned}
\tag{6}
$$

$$+ \quad X_1\left(\sqrt[3]{X_0 - 2345.71} - X_2\right) + \sqrt{\cos(X_2)} + X_4) + X_0,$$

$$
\begin{aligned}
y_5 \quad = \quad & - \max(7.64(X_1 + X_3), 0.43\log(\min((X_2|\min(221.844X_2\cot(\cos(X_2)), -9574.83 \\
& \min(8546.04(X_1 + X_3), X_8(X_0 + X_1 + \min((2X_0 + X_1)X_8, 1529.14(1529.14(X_1 + X_{17}) \\
& - \quad (2X_0 + X_1)X_8))))))|\Big)\Big/\Big(|X_1 + \sqrt[3]{|\cos(X_2)|}|\Big), 0.43\log(\sqrt[9]{\sin(\max(\sqrt{X_0}, \sqrt[3]{X_4}))})))) \\
& + \quad |\min(7.33(X_1 + X_3), X_8(X_0 + X_1 + |\cos(X_2)|))| - 21.09.
\end{aligned}
\tag{7}
$$

However, it should be noted that, to use Equations (3)–(7) the values of input variables must be scaled using the StandardScaler method. In Equations (3)–(7), the mathematical functions such as the square root, natural logarithm, logarithm with base 2 and 10, and division function are specifically defined to avoid complex, infinite values or zero division errors. The definition of these functions is given in Appendix A.

The analysis of Equations (3)–(7) showed that, using all five SEs together to compute the output all 19 variables is required. However, if Equations (3)–(7) are used individually, not all input variables are required. To compute the output in Equation (3), 13 out of 19 input variables are required. The variables that are required to compute the output are $X_0,...,X_9$, $X_{15}$, $X_{17}$, and $X_{18}$. It is evident from Table 2 that these variables are PBstart, PCstart, PDstart, PFstart, PArise, PBrise, PCrise, PDrise, PFrise, PAfall, PBwidth, PCwidth, and PFwidth, respectively. The total number of variables required to compute the output using Equation (4) is 10 out of 19, and these variables are $X_1,...,X_6$, $X_8$, $X_{14}$, $X_{15}$, and $X_{18}$. From Table 2, these input variables are PCstart, PDstart, PFstart, PArise, PBrise, PCrise, PFrise, PAwidth, PBwidth, and PFwidth, respectively. In Equation (5), to compute the output, the 12 out of 19 input variables required are $X_0,...$, $X_8$, $X_{14}$, $X_{16}$, and $X_{18}$. These variables are PBstart, PCstart, PDstart, PFstart, PArise, PBrise, PCrise, PDrise, PFrise, PAwidth, PCwdith, and PFwidth, respectively. All input variables are required to compute the output using Equation (6). The lowest number of input variables (7) is required to compute the output when Equation (7) is used. Equation (7) consist of $X_0,...,X_4$, $X_8$, and $X_{17}$, and from Table 2, these variables are PBstart, PCstart, PDstart, PFstart, PArise, PFrise, and PDwidth.

The set of previously shown SEs was evaluated on the entire dataset, and the results are presented in Figure 8 and Table 8.

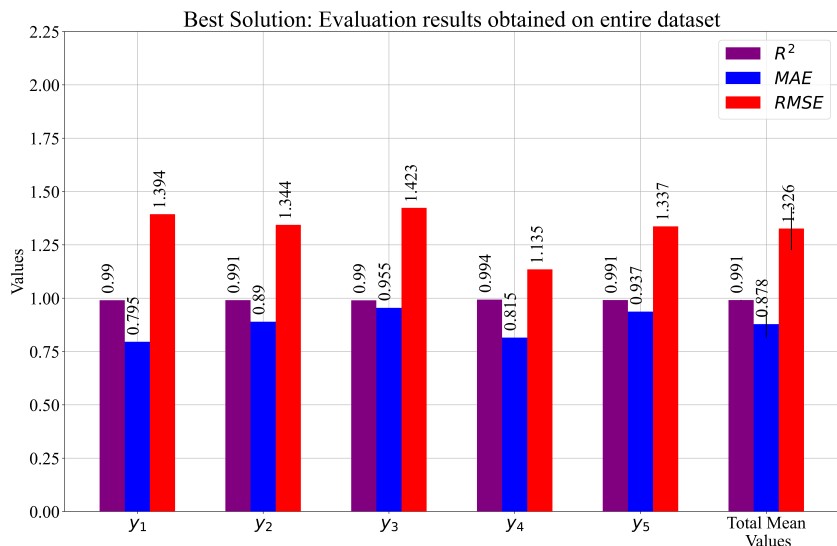

**Figure 8.** The graphical representation of the performance metric values obtained with the best combination of SEs on the entire dataset. $\sigma$ is shown in the case of the total mean values in the form of an error bar.

**Table 8.** The performance metric values achieved with the best combination of SEs on the entire dataset.

| Evaluation Metric | $y_1$ | $y_2$ | $y_3$ | $y_4$ | $y_5$ | Mean | $\sigma$ |
|---|---|---|---|---|---|---|---|
| $R^2$ | 0.99022 | 0.990905 | 0.989798 | 0.993511 | 0.991003 | 0.991087 | 0.001291 |
| $MAE$ | 0.795499 | 0.889602 | 0.954781 | 0.815055 | 0.936876 | 0.878362 | 0.063662 |
| $RMSE$ | 1.393535 | 1.343819 | 1.423247 | 1.135059 | 1.336586 | 1.326449 | 0.100901 |

The results shown in Table 8 showed that, using a customized combination of the SEs, even higher estimation accuracy was achieved. The mean values were better than those shown in Table 7, and the $\sigma$s were smaller.

The reconstruction of the interaction locations was performed in [13], where the authors used the DNN. For the evaluation of the results, the authors used the $RMSE$, and the lowest achieved value was 1.53. If these results are compared to the results shown in Table 8, it can be seen that the proposed method outperformed the results in [13].

## 4. Discussion

The initial statistical analysis of the dataset variables showed that all the input variables had a really small value range when compared to the output (target) variable. With this in mind, scaling actions were taken, i.e., all input variables were scaled using the Standard-Scaler method. Based on Pearson's correlation analysis, it was possible to see that the (6 out of 18) input variables had a very good correlation with the output variable $y$. Only one (PBStart) dataset variable did not correlate with the output variable.

Although the initial investigation of the GPSR algorithm to define each GPSR hyperparameter range for the development of the RHS method was a painstaking and time-consuming process, eventually, it resulted in the faster search of multiple hyperparameter combinations, for which the highest estimation accuracies were achieved. The larger population size and smaller tournament selection size proved to be a good combination in obtaining the SEs with high evaluation accuracy; however, these two were not only responsible. An initial investigation showed that, initially, a small range of any genetic operation would lead to a local minimum value of the fitness value, which would eventually result in SEs with lower estimation accuracy. To improve the range of all genetic operations, it was set to the 0.001–1 range. This wide range produced some interesting hyperparameter combinations (Table 4). In Case 1, the dominant genetic operation was point mutation (0.43). Cases 2 and 3 had high values of crossover and subtree mutation, i.e., they were the main genetic operations in these two cases. As already stated, the idea was to enable the GPSR algorithm to reach the lowest value of the fitness function possible, so the stopping criteria were preset to an extremely low value, and the GPSR execution ended after a maximum number of generations was reached.

When Case 1's hyperparameters and the SEs' lengths were compared to those of Cases 2 and 3, it can be noticed that the initial tree depth size and Pcoef had a huge influence on the length and depth of the SEs. Almost all the SEs obtained in Case 1 were large in length and depth. The lowest SE size was obtained on "Split 3" with a length of 106 and a depth of 13. However, these SEs had the lowest estimation performance on the test dataset ($R^2 = 0.9811$, $MAE = 1.1654$, and $RMSE = 1.9488$) when compared to the other four SEs of the same case. Generally, the Pcoef in Case 1 was set to $9.15 \times 10^{-6}$, which generated SEs of average length 206.6, as seen from Table 6. The Pcoef in Cases 2 and 3 was much larger ($9.2 \times 10^{-4}$, $7.16 \times 10^{-4}$), which generated average lengths of 81.4 and 161.6, respectively. From these results, the high influence of the Pcoef on the size of the obtained SEs can be noticed.

From the results obtained on the training and testing datasets (Figures 6 and 7 and Tables 5 and 7), it can be seen that the mean values of the performance metrics were similar (values of $R^2$ high and near 1, while $MAE$ and $RMSE$ values of 1 and 1.5, respectively). However, in Case 2, the performance metric values were slightly higher on the training

compared to the testing dataset. Therefore, Case 3 slightly outperformed Case 2 on the testing dataset. From the obtained results, the highest evaluation metric values ($\overline{R^2} \pm \sigma(R^2)$, $\overline{MAE} \pm \sigma(MAE)$, $\overline{RMSE} \pm \sigma(RMSE)$) on the testing dataset were achieved in Case 1 (0.9876 ± 0.0048, 1.0096 ± 0.1215, 1.5506 ± 0.3050.)

The combined set of the five best SEs selected from Cases 1, 2, and 3 achieved the highest estimation accuracy when evaluated on the entire dataset. The results of the evaluation metrics ($\overline{R^2} \pm \sigma(R^2)$ and $\overline{MAE} \pm \sigma(MAE)$, $\overline{RMSE} \pm \sigma(RMSE)$) in this case were equal to 0.991087 ± 0.001291, 0.878362 ± 0.063662, and 1.326449 ± 0.100901. Therefore, the results showed that the custom combination of the best SEs in terms of the evaluation metric values outperformed Cases 1–3.

The analysis of the number of variables required to compute the output if a custom solution (five SEs) is used showed that all input variables were required. Even if only Equation (5) was used, all input variables were required. However, the high estimation accuracy can be achieved only using Equation (7), and this equation required only seven input variables. However, 4 (PCstart, PFstart, PArise, and PFrise) out of those 7 variables had a high correlation with the output variable, as seen from Figure 3.

## 5. Conclusions

In the conducted research, the GPSR algorithm with RHS and five-fold CV was used to obtain a system of robust SEs for the estimation of the interaction locations in Super Cryogenic Matter Search detectors. The results of the investigation showed that:

- Using the GPSR algorithm, it was possible to obtain SEs (mathematical equations) that can estimate the interaction locations in Super Cryogenic Matter Search detectors with high accuracy.
- Using a 5-CV process, the robust system of the five SEs has a more accurate estimation when compared to the estimation of only one SE. The RHS method proved to be very useful in finding the combination of the quality hyperparameters where the highest accuracy was achieved. From all the results, the best three cases of the SEs were selected and evaluated on the test set. The highest values of $\overline{R^2} \pm \sigma(R^2)$, $\overline{MAE} \pm \sigma(MAE)$, $\overline{RMSE} \pm \sigma(RMSE)$ were achieved in Case 1 and are equal to 0.9876 ± 0.0048, 1.0096 ± 0.1215, 1.5506 ± 0.3050.
- From the obtained results, three cases were selected based on the final mean $R^2$ score. From these cases, the SEs that achieved the highest estimation performance were selected as the main elements of the custom set of SEs. The results of the customized solution obtained on the entire dataset were equal to 0.991087 ± 0.001291, 0.878362 ± 0.063662, and 1.326449 ± 0.100901, respectively. The final evaluation of these equations on the entire dataset showed that this system had slightly better performance when compared to Cases 1, 2, and 3.
- Unfortunately, the custom set of SEs required all 19 input variables to compute the output. However, if only Equation (7) was used, the highest estimation accuracy could be achieved, and to compute the output, only seven input variables were required.

The proposed approach presented in this paper showed that using a simple GPSR algorithm with RHS and the 5-CV on low-end computer hardware can produce better results than the complex CNN architecture presented in [13]. The benefit of using the proposed approach is that the SEs were easily used, easy to comprehend, and require fewer computational resources than complex CNN architectures.

The main problem of the proposed approach is the initial definition of the GPSR hyperparameter ranges in the RHS method. The ranges are not unique and depend on the investigation, so each time, it has to be tuned from scratch. Depending on the dataset, the population size, number of generations, and tournament selection have to be defined. The larger the dataset size, the smaller the population, number of generations, and tournament selection size values are. Besides those hyperparameters, the Pcoef has to be defined, and this parameter is the most-sensitive one. A small increase/decrease of this value can result in a fast increase/decrease in the length of the population members.

The other important factor that influences the hyperparameter ranges is the computational resources on which the GPSR algorithm with the RHS method is executed.

Future work will be focused on synthetically enlarging the dataset size to see if the estimation accuracy could be improved. Besides that, other ML methods will be investigated, especially ensemble methods, with the idea of improving the performance metric values as much as possible.

**Author Contributions:** Conceptualization, N.A. and S.B.Š.; methodology, Z.C.; software, N.A. and M.G.; validation, N.A., S.B.Š., and M.G.; formal analysis, N.A., S.B.Š., and M.G.; investigation, N.A., S.B.Š., and M.G.; resources, N.A., M.G., and Z.C.; data curation, N.A., M.G., and Z.C.; writing—original draft preparation, N.A., S.B.Š., and M.G.; writing—review and editing, N.A., S.B.Š., and M.G.; visualization, N.A., S.B.Š., and M.G.; supervision, N.A. and Z.C.; project administration, Z.C.; funding acquisition, Z.C. All authors have read and agreed to the published version of the manuscript.

**Funding:** This research received no external funding.

**Institutional Review Board Statement:** Not applicable.

**Informed Consent Statement:** Not applicable.

**Data Availability Statement:** The data used in this paper was obtained from a publicly available repository located at https://www.kaggle.com/datasets/fairumn/cdms-dataset, accessed on 9 January 2023.

**Acknowledgments:** This research was (partly) supported by the CEEPUS network CIII-HR-0108, the European Regional Development Fund under Grant KK.01.1.1.01.0009 (DATACROSS), the project CEKOM under Grant KK.01.2.2.03.0004, the Erasmus+ project WICT under Grant 2021-1-HR01-KA220-HED-000031177, and the University of Rijeka Scientific Grants uniri-mladi-technic-22-61 and uniri-tehnic-18-275-1447.

**Conflicts of Interest:** The authors declare no conflict of interest.

## Appendix A. Additional Description of Mathematical Functions in SEs

In GPSR, some mathematical functions such as the square root, natural logarithm, logarithm with base 2 and 10, and division are differently defined to avoid generating imaginary or inf values. To avoid imaginary values, these functions have to be applied when the SEs defined with Eqs. (3)-(7) are used. The mathematical functions are defined as follows:

- Square root:

$$y_{sqrt}(x) = \sqrt{|x|}, \tag{A1}$$

- Natural logarithm:

$$y_{\log}(x) = \begin{cases} \log(|x|) & \text{if } |x| > 0.001 \\ 0 & \text{if } |x| < 0.001 \end{cases}, \tag{A2}$$

and when natural logarithms with base 2 and 10 are used, the log in Equation (A2) is replaced with $\log_2$ or $\log_{10}$, respectively.

- Division:

$$y_{div}(x_1, x_2) = \begin{cases} \frac{x_1}{x_2} & \text{if } |x_2| > 0.001 \\ 1 & \text{if } |x_2| < 0.001 \end{cases}. \tag{A3}$$

The variables $x$, $x_1$, and $x_2$ in Equations (A1 - A3) are arbitrary values and have no relation with the input dataset variables.

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
