# Peer review of "Estimation of Interaction Locations in Super Cryogenic Dark Matter Search Detectors Using Genetic Programming-Symbolic Regression Method"

_applsci, doi:10.3390/app13042059_

Round 1

Reviewer 1 Report

The paper introduces a food technique for estimating interaction locations using genetic programming regression.

The paper should be modified to increase its contribution. Here are some comments:

1- Grammar and langugage require extensive revision. Some highlights are shown in the attached PDF.

2- Abstract contains detailed/numerical results which should be moved to discussion and conclusion sections.

3- In section 1.2, bullets should start with capital letters.

4- The symbolic formulas contain square roots of quantities that may be negative and may produce errors. Some verifications should be added.

5- Some equations (e.g. equation 4) are well known and no need to write the details.

6- Reference # 13 is not identified enough. More details should be added.

Author Response

The authors want to thank the reviewer for his comments and suggestions which have greatly improved the manuscript's quality. The authors do hope that the manuscript will be considered for publication. The answers to the reviewer's comments are given below. 

The paper introduces a food technique for estimating interaction locations using genetic programming regression.

The paper should be modified to increase its contribution. Here are some comments:

1- Grammar and langugage require extensive revision. Some highlights are shown in the attached PDF.

Answer: The authors want to thank the reviewer for highlighting the errors in the manuscript. To the best of our knowledge, the grammar and language errors have been corrected in the entire manuscript. It should be noted that when the revised version of the manuscript was completed the manuscript was corrected with the use of Grammarly.  The correction of grammatical and language errors which the reviewer highlighted in the document entitled “peer-review-26769434.v1.pdf” are shown below. 

  1. Bullets in subsection 1.2 should start with capital letters (This is the 3rd comment of the same reviewer)- All bullets now start with capital letters as shown in a revised version of the manuscript. 
  2. The first two words of the paragraph after the bullets are highlighted. 
    1. Citing from the original version of the manuscript: “Presented paper consists of the following sections …
    2. Citing from the revised version of the manuscript:  “The presented paper consists of the following sections … 
  3. The last sentence of the paragraph below the section “Materials and Methods”. 
    1. Citing from the original version of the manuscript: “Finally, the computational resources are described using which the proposed method was executed.” 
    2. Citing from the revised version of the manuscript: “Finally, the computational resources used in this research, are described.”
  4. The first sentence in the paragraph below the subsection entitled “Dataset description”. 
    1. Citing from the original version of the manuscript: “For this investigation, the publicly available dataset from Kaggle was used [12].” 
    2. Citing the revised version of the manuscript: “For this investigation, the publicly available dataset from Kaggle [12], was used.”

2- Abstract contains detailed/numerical results which should be moved to discussion and conclusion sections.

Answer: The final part of the abstract where detailed/numerical results are presented is moved to the discussion and the conclusion section. 

Citing abstract from the revised version of the manuscript: 

“The Super Cryogenic Dark Matter Search (SuperCDMS) experiment is used to search for Weakly Interacting Massive Particles (WIMPs) - candidates for dark matter particles. In this experiment the WIMPs interact with nuclei in the detector however, there are a lot of other interactions (background interactions). To separate background interactions from the signal it is necessary to measure the interaction energy and to reconstruct the location of interaction between WIMPs and the nucleus. In recent years some research papers are investigating the reconstruction of interaction locations using artificial intelligence (AI) methods.  In this paper, a genetic programming-symbolic regression (GPSR), with randomly tuned hyperparameters cross-validated via a 5-fold procedure was applied to the SuperCDMS experiment to estimate the interaction locations with high accuracy. To measure the estimation accuracy of obtaining (SE) the mean and standard deviation ($\sigma$) values of $R^2$, root mean squared error ($RMSE$), and finally, the mean absolute error ($MAE$) were used. \color{cyan}The investigation showed that using GPSR SEs can be obtained that estimates the interaction locations with high accuracy. To improve the solution the 5 best SEs were combined from the 3 best cases. Results demonstrate that a very high estimation accuracy can be achieved with the proposed methodology.\color{black}”

Citing Discussion section from the revised version of the manuscript: 

The initial statistical analysis of dataset variables showed that all the input variables have a really small value range when compared to the output (target) variable. With this in mind, scaling actions were taken, i.e. all input variables were scaled using the Standard Scaler method. Based on Pearson's correlation analysis, it was possible to see that (6 out of 18) input variables have a very good correlation with the output variable $y$. Only one (PBStart) dataset variable does not correlate with the output variable. \newline 

Although the initial investigation of the GPSR algorithm to define each GPSR hyperparameter range for the development of the RHS method was a painstaking and time-consuming process eventually it resulted in the faster search of multiple hyperparameter combinations using which the highest estimation accuracies were achieved. The larger population size and smaller tournament selection size proved to be a good combination in obtaining SEs with high evaluation accuracy however, these two were not only responsible. Initial investigation showed that initially, a small range of any genetic operation would lead to a local minimum value of fitness value which would eventually result in SEs with lower estimation accuracy. To improve that, the range of all genetic operations was set to 0.001-1 range. This wide range produced some interesting hyperparameter combinations (Table \ref{tab:BestHyperParmeters}). In case 1 the dominating genetic operation was point mutation (0.43). Cases 2 and 3 had high values of crossover and subtree mutation i.e. they were to main genetic operations in these two cases.  As already stated the idea was to enable the GPSR algorithm to reach the lower value of the fitness function possible so the stopping criteria were preset to an extremely low value and the GPSR execution ended after a maximum number of generations was reached.\newline  When Case 1 hyperparameters and SEs lengths are compared to those of Cases 2 and 3 it can be noticed that the initial tree depth size and Pcoef had a huge influence on the length and depth of SEs. Almost all SEs obtained in Case 1 were large in length and depth. The lowest SE size was obtained on "Split 3" with a length of 106 and a depth of 13. However, these SEs had the lowest estimation performance on the test dataset ($R^2 = 0.9811$, $MAE = 1.1654$, and $RMSE = 1.9488$) when compared to the other four SEs of the same case. Generally, the Pcoef in case 1 was set to $9.15\times 10^{-6}$ which generated SEs of average length 206.6  as seen from Table \ref{tab:SymbolicSizeTime}. The Pcoef in cases 2 and 3 were much larger ($9.2\times 10^{-4}$, $7.16\times 10^{-4}$) which generated average lengths of 81.4 and 161.6, respectively. From these results, it can be noticed a high influence of the Pcoef on the size of the obtained SEs.     \newline 

From results obtained on training and testing datasets (Figures \ref{fig:5CVGraphic}, \ref{fig:TestEvalGraphic} and Tables \ref{tab:5CVNumeric}, \ref{tab:evalTest}) it can be seen that mean values of performance metrics are similar (values of $R^2$ high and near 1 while $MAE$ and $RMSE$ values equal to 1 and 1.5 respectively). However, in Case 2 the performance metric values are slightly higher on the training than on the testing dataset. So, Case 3 slightly outperforms Case 2 on the testing dataset. \color{cyan} From obtained results, the highest evaluation metric values ($\overline{R^2} \pm \sigma(R^2)$, $\overline{MAE} \pm \sigma(MAE)$, $\overline{RMSE}\pm \sigma(RMSE)$) on the testing dataset where achieved in Case 1 ($0.9876 \pm 0.0048$, $1.0096 \pm 0.1215$, $1.5506 \pm 0.3050$.)\color{black}\newline

\color{cyan}The combined set of 5 best SEs selected from Cases 1,2, and 3 achieved highest estimation accuracy when evaluated on the entire dataset. The results of evaluation metrics ($\overline{R^2} \pm \sigma(R^2)$ , $\overline{MAE} \pm \sigma(MAE)$, $\overline{RMSE}\pm \sigma(RMSE)$) in this case are equal to $0.991087 \pm 0.001291$, $0.878362 \pm 0.063662$, and $1.326449 \pm 0.100901$. So, the results showed that the custom combination of the best SEs in terms of evaluation metric values outperformed Cases 1-3. \newline\color{black}

The analysis of the number of variables required to compute the output if a custom solution (5 SEs) is used, all input variables are required. Even if only Eq. (\ref{eq:besteq3}) is used all input variables are required. However, the high estimation accuracy can be achieved only using Eq.(\ref{eq:besteq5}) and this equation requires only 7 input variables. However, 4 (PCstart, PFstart, PArise, and PFrise) out of those 7 variables have a high correlation with the output variable as seen from Figure \ref{fig:PHeatMap}.

Citing Conclusions section from the revised version of the manuscript:

In the conducted research, the GPSR algorithm with RHS and 5-fold CV was used to obtain a system of robust SEs for estimation of the interaction locations in super cryogenic matter search detectors. The results of the investigation showed: 

\begin{itemize}

\item using GPSR algorithm it was possible to obtain SEs (mathematical equations) that can estimate the interaction locations in Super Cryogenic Matter Search detectors with high accuracy, 

\item using a 5-CV process the robust system of 5 SEs which estimation is more accurate when compared to the estimation of only one SE. The RHS method proved to be very useful in finding the combination of quality hyperparameters where the highest accuracy was achieved. \color{cyan}From all the results the best three cases of SEs were selected and evaluated on the test set. The highest values of $\overline{R^2} \pm \sigma(R^2)$, $\overline{MAE} \pm \sigma(MAE)$, $\overline{RMSE}\pm \sigma(RMSE)$ were achieved in Case 1 and are equal to $0.9876 \pm 0.0048$, $1.0096 \pm 0.1215$, $1.5506 \pm 0.3050$.\color{black}

\item from obtained results three cases were selected based on the final mean $R^2$ score. From these cases, the SEs which achieved the highest estimation performance was selected as the main elements of the custom set of SEs. \color{cyan}The results of the customized solution obtained on the entire dataset are equal to $0.991087 \pm 0.001291$, $0.878362 \pm 0.063662$, and $1.326449 \pm 0.100901$, respectively. \color{black} The final evaluation of these equations on the entire dataset showed that this system has slightly better performance when compared to Cases 1,2 and 3. 

\item unfortunately the custom set of SEs requires all 19 input variables to compute the output. However, if only Eq.(\ref{eq:besteq5}) is used the highest estimation accuracy could be achieved, and to compute the output only 7 input variables are required.

\end{itemize}

The proposed approach presented in this paper showed that using a simple GPSR algorithm with RHS and 5-CV on low-end computer hardware can produce better results than complex CNN architecture presented in \cite{fair2021doc}. The benefit of using the proposed approach is SEs that are easily used, easy to comprehend, and require fewer computational resources than complex CNN architectures. \newline 

The main problem of the proposed approach is the initial definition of GPSR hyperparameter ranges in the RHS method. The ranges are not unique and depend on the investigation so each time it has to be tuned from scratch. Depending on the dataset, population size, number of generations, and tournament selection have to be defined. The larger the dataset size the smaller population, number of generations, and tournament selection size values are. Besides those hyperparameters, the Pcoef has to be defined and this parameter is the most sensitive one. A small increase/decrease of this value can result in a fast increase/decrease in the length of population members. The other important factor which influences the hyperparameter ranges are computational resources on which the GPSR algorithm with the RHS method is executed.  \newline 

Future work will be focused on synthetically enlarging the dataset size to see if the estimation accuracy could be improved. Besides, that other ML methods will be investigated especially ensemble methods with the idea of improving the performance metric values as much as possible. \newline

3- In section 1.2, bullets should start with capital letters.

Answer: In the revised version of the manuscript the bullets start with capital letters. 

Citing from the revised version of the manuscript: 

In comparison to the reviewed research, research questions can be posed as follows:

\begin{itemize}

\item Can a SE be obtained using GPSR which can accurately reconstruct the locations of interactions in SuperCDMS detectors?

\item Is it possible to obtain a set of robust SEs using GPSR with randomly tested hyperparameters and validated through k-fold cross-validation that can reconstruct the locations of interactions in the SuperCDMS with high accuracy?

\item Is it possible to achieve even higher estimation accuracy in the reconstruction of the locations interactions by combining multiple SEs that were obtained from different GPSR executions?

\item Whether all input variables are required as model inputs to accurately reconstruct interaction locations?

\end{itemize}

\color{black}

4- The symbolic formulas contain square roots of quantities that may be negative and may produce errors. Some verifications should be added.

Answer: The authors want to apologize for not clarifying the usage of the symbolic expression. The GPSR treats some mathematical functions differently: square root, natural logarithm, and logarithm with bases 2 and 10. The mathematical functions are defined in this way to avoid generating imaginary values or infinity in some cases which can obviously occur if the math functions are defined without these restrictions. So in the case of square root the argument of the function whether positive or negative the absolute value is obtained and then the square root is applied. So in any case it cannot produce an error.  The usage of these mathematical functions is clearly presented in appendix A. 

The reference to the appendix is placed in the GPSR algorithm description to clearly indicate that the used mathematical functions are specifically defined. Citing the additional description of these mathematical functions from the section entitled “” from the revised manuscript version: 

5- Some equations (e.g. equation 4) are well known and no need to write the details.

Answer: The authors agree with suggestion and these formulas were omitted from the revised version of the manuscript. Only a short description was given in the subsection 2.6. Citing from the revised version of the manuscript: 

In this paper to evaluate SEs, three performance metrics were used i.e. the coefficient of determination (R^2) [25], the mean absolute error $MAE$ [18] (Eq.(2)), and the root mean squared error (RMSE) [26]. The R^2 is in the 0 to 1 range where 0 represents the worst possible value while the value 1 represents the best possible value and is aimed for. Regarding MAE, and RMSE values the goal is to obtain as low as possible values.

6- Reference # 13 is not identified enough. More details should be added.

Answer: The authors agree with the lack of information about reference #13. The reference describes the initial investigation done by the authors of the dataset and is available on the GitHub repository. In the revised version of the manuscript, reference #13 was updated with the link to the paper available at GitHub repository. 

The Link to the paper is available on the GitHub repository: https://github.com/FAIR-UMN/FAIR-UMN-CDMS/blob/main/doc/FAIR%20Document%20-%20Identifying%20Interaction%20Location%20in%20SuperCDMS%20Detectors.pdf

Reviewer 2 Report

In this paper, a genetic programming-symbolic 7 regression (GPSR), with randomly tuned hyperparameters cross-validated via a 5-fold procedure 8 was applied to the SuperCDMS experiment to estimate the interaction locations with high accuracy.This is a real world problem and makes sense, but I feel there are a few issues that need improvement or clarification:

1 Why did you improve the algorithm in this way?  What is the biggest flaw in the current algorithm? And is there any significant improvement between the calculation results of your algorithm and the current algorithm? You need to spell it out.

2 Can you compare your method with the most common other methods to illustrate the superiority of your algorithm?

3 Can you use the benchmark function to verify the performance of this algorithm.

Author Response

In this paper, a genetic programming-symbolic 7 regression (GPSR), with randomly tuned hyperparameters cross-validated via a 5-fold procedure 8 was applied to the SuperCDMS experiment to estimate the interaction locations with high accuracy. This is a real world problem and makes sense, but I feel there are a few issues that need improvement or clarification:

1 Why did you improve the algorithm in this way?  What is the biggest flaw in the current algorithm? And is there any significant improvement between the calculation results of your algorithm and the current algorithm? You need to spell it out.

Answer: The authors have made a clear indication of the novelty in the original version of the manuscript. The short literature review showed that the majority of used AI algorithms in dark matter research gravitate to the use of Deep neural networks or convolutional neural networks. Although these algorithms are excellent in the majority of detecting or estimating the problem with these algorithms is that they are computationally intensive and the train models cannot be transformed into a simple mathematical equation that connects the input with the output. 

The proposed method used in this research is an evolutionary algorithm GPSR that generates a mathematical equation that connects input with output variables. No matter how long the equation is, the equation is easy to integrate and use and does not require significant computational resources to compute the output. 

Citing from the original and the revised version of the manuscript (novelty in the introduction section): “The authors present a novel approach through the use of genetic programming symbolic regression (GPSR) that was applied to determine equations that can accurately reconstruct the particle interaction locations in the SuperCDMS. A public dataset \cite{fairumn2021}  provided by a team from the University of Minnesota has been used in the research.

The advantages and disadvantages of the proposed method used in this research are given at the end of the Conclusions section of the original and revised version of the manuscript. 

Citing from the original and revised version of the manuscript (advantages of the proposed method): “The proposed approach presented in this paper showed that using a simple GPSR algorithm with RHS and 5-CV on low-end computer hardware can produce better results than complex DNN architecture presented in \cite{fair2021doc}. The benefit of using the proposed approach is SEs that are easily used, easy to comprehend and require fewer computational resources than complex DNN architectures.

Citing from the original and revised version of the manuscript (disadvantages of the proposed method): “The main problem of the proposed approach is the initial definition of GPSR hyperparameter ranges in the RHS method. The ranges are not unique and depend on the investigation so each time it has to be tuned from scratch. Depending on the dataset, population size, number of generations, and tournament selection have to be defined. The larger the dataset size the smaller population, number of generations, and tournament selection size values are. Besides those hyperparameters, the Pcoef has to be defined and this parameter is the most sensitive one. A small increase/decrease of this value can result in a fast increase/decrease in the length of population members. The other important factor which influences the hyperparameter ranges are computational resources on which the GPSR algorithm with the RHS method is executed.

2 Can you compare your method with the most common other methods to illustrate the superiority of your algorithm?

Answer: We have compared the results obtained with this method with research in which the DNN was used to obtain the output. Since this is the only research, to the best of our knowledge, conducted on this dataset. 

Citing from the original and revised version of the manuscript (the last paragraph of the subsection Combination of the best SE): 

“The reconstruction of the interaction locations was done in \cite{fair2021doc} where authors have used the DNN. For the evaluation of the results the authors have used $RMSE$ and the lowest achieved value was 1.53. If these results are compared to the results shown in Table \ref{tab:BestAll} it can be seen that the proposed method outperforms the results in \cite{fair2021doc}.

3 Can you use the benchmark function to verify the performance of this algorithm.

Answer: We are not quite sure what the reviewer meant by the “benchmark function” so we will try to describe the proposed methodology used in this paper. The GPSR algorithm is an evolutionary algorithm that requires the dataset in which the input and output variables are defined to generate the initial population of symbolic expressions (mathematical equations) as well as to generate the final “best” symbolic expression. To generate the initial population and the final “best” symbolic expression the GPSR algorithm requires the definition of the range of constants and the mathematical functions among many other hyperparameters. 

The “benchmark function” for this problem does not exist since the novelty of this paper was to obtain symbolic expression specifically for the detection of interaction locations in the SuperCDMS detector. In the other research in which this was also investigated (reference 13 in the original and revised version of the manuscript), the authors used DNN and the best result obtained in terms of RMSE was achieved with DNN-2 (2 hidden layers) where the training set the RMSE was equal to 1.53, on Validation set 1.533, and Test set 1.741.  In this case, the trained DNN model is obtained which cannot easily be transformed into a symbolic expression and requires more computational resources than a simple symbolic expression. 

The benefit of using our approach is the 5 symbolic expressions which achieved a mean RMSE value of 1.3264 with a standard deviation of 0.1 (see Table 8).

Reviewer 3 Report

The authors presented a symbolic genetic algorithm search with a hyperparameter search to estimate the interaction locations in super cryogenic matter search detectors. In the introduction of the papers, the authors show the motivation and the actuality of the research, which is a current problem. The next part shows the other machine learning-based methodologies used to resolve the proposed problem then the final part of the introduction shows the novelty of the paper point-by-point. The authors also pointed out that they used an openly accessible kaggle benchmark problem to test and validate their methodology. In the methodology chapter they show their approach point by point. The results are discussed in detail in a separate section. I have no major remarks with this paper.

Author Response

The authors want to thank the reviewer for his time and effort to give constructive comments and suggestions which have greatly improved the manuscript's quality. The authors do hope that manuscript in this form will be considered for publication. The answer to the reviewers comments is given below. 

The authors presented a symbolic genetic algorithm search with a hyperparameter search to estimate the interaction locations in super cryogenic matter search detectors. In the introduction of the papers, the authors show the motivation and the actuality of the research, which is a current problem. The next part shows the other machine learning-based methodologies used to resolve the proposed problem then the final part of the introduction shows the novelty of the paper point-by-point. The authors also pointed out that they used an openly accessible kaggle benchmark problem to test and validate their methodology. In the methodology chapter, they show their approach point by point. The results are discussed in detail in a separate section. I have no major remarks with this paper.

Answer: The authors want to thank the reviewer for reviewing this manuscript and for his positive comments.

Round 2

Reviewer 1 Report

Ok

The recommendations have been considered. 

Reviewer 2 Report

It has been properly revised and can be employed